# Lung and Gut Microbiota Changes Associated with *Pseudomonas aeruginosa* Infection in Mouse Models of Cystic Fibrosis

**DOI:** 10.3390/ijms222212169

**Published:** 2021-11-10

**Authors:** Giovanni Bacci, Alice Rossi, Federica Armanini, Lisa Cangioli, Ida De Fino, Nicola Segata, Alessio Mengoni, Alessandra Bragonzi, Annamaria Bevivino

**Affiliations:** 1Department of Biology, University of Florence, Sesto Fiorentino, 50019 Florence, Italy; giovanni.bacci@unifi.it (G.B.); lisa.cangioli@unifi.it (L.C.); alessio.mengoni@unifi.it (A.M.); 2Infections and Cystic Fibrosis Unit, Division of Immunology, Transplantation and Infectious Diseases, IRCCS San Raffaele Scientific Institute, 20132 Milan, Italy; rossi1.alice@hsr.it (A.R.); defino.ida@hsr.it (I.D.F.); bragonzi.alessandra@hsr.it (A.B.); 3Department CIBIO, University of Trento, 38122 Trento, Italy; federica.armanini@unitn.it (F.A.); nicola.segata@unitn.it (N.S.); 4Department for Sustainability, Italian National Agency for New Technologies, Energy and Sustainable Economic Development, ENEA Casaccia Research Center, 00123 Rome, Italy

**Keywords:** cystic fibrosis, lung, gut, microbiome, gut-lung axis, animal models, CFTR mice, *Pseudomonas aeruginosa*

## Abstract

Cystic fibrosis (CF) disease leads to altered lung and gut microbiomes compared to healthy subjects. The magnitude of this dysbiosis is influenced by organ-specific microenvironmental conditions at different stages of the disease. However, how this gut-lung dysbiosis is influenced by *Pseudomonas aeruginosa* chronic infection is unclear. To test the relationship between CFTR dysfunction and gut-lung microbiome under chronic infection, we established a model of *P. aeruginosa* infection in wild-type (WT) and gut-corrected CF mice. Using 16S ribosomal RNA gene, we compared lung, stool, and gut microbiota of C57Bl/6 *Cftr ^tm1UNC^*TgN(FABP*CFTR*) or WT mice at the naïve state or infected with *P. aeruginosa.* *P. aeruginosa* infection influences murine health significantly changing body weight both in CF and WT mice. Both stool and gut microbiota revealed significantly higher values of alpha diversity in WT mice than in CF mice, while lung microbiota showed similar values. Infection with *P. aeruginosa* did not changed the diversity of the stool and gut microbiota, while a drop of diversity of the lung microbiota was observed compared to non-infected mice. However, the taxonomic composition of gut microbiota was shown to be influenced by *P. aeruginosa* infection in CF mice but not in WT mice. This finding indicates that *P. aeruginosa* chronic infection has a major impact on microbiota diversity and composition in the lung. In the gut, CFTR genotype and *P. aeruginosa* infection affected the overall diversity and taxonomic microbiota composition, respectively. Overall, our results suggest a cross-talk between lung and gut microbiota in relation to *P. aeruginosa* chronic infection and CFTR mutation.

## 1. Introduction

Cystic fibrosis (CF) is the most common autosomal recessive disease caused by mutations in the Cystic Fibrosis Transmembrane Conductance Regulator (*CFTR*) gene [1]. Dysfunction of *CFTR* affects several organs, but primarily complications in the lung are the causes of morbidity and mortality [2]. Absence of CFTR-mediated Cl-secretion has been functionally linked to airway surface dehydration, which leads to the accumulation of concentrated mucus, airway obstruction, inflammation and infection, bronchiectasis, and ultimately death. Gastrointestinal (GI) manifestations are the most important non-pulmonary manifestations of CF and commonly complicate the care of CF patients. GI tract manifestations of CF are related to mucous inspissation and dysmotility and include meconium ileus, constipation, distal intestinal obstruction syndrome, gastroesophageal reflux disease, and small bowel bacterial overgrowth. In CF, the central role of lung and gut microbiomes in the pathophysiology has been supported by several studies [3]. CF patients show altered airway microbiome, correlated to lung-specific microenvironmental conditions (e.g., viscous mucus, hyperinflammation, and infection) and altered fecal microbiomes, correlated with gastrointestinal inflammation and nutrient malabsorption [3,4]. Emerging findings show a link between gut microbiome and progression of lung disease in CF [5,6] similar to other lung diseases such as asthma, COPD, allergy, and lung cancer [7,8,9]. The altered composition of the gut microbiome has been shown to profoundly impact the lung inflammatory response in the context of pulmonary diseases via the vital cross-talk between the lungs and gut microbiota [10,11,12]. Evidence that CFTR mutations have an effect on gut microbiota are increasing in humans and in CF mice [13,14], although it is unclear how this gut-lung dysbiosis is influenced by *Pseudomonas aeruginosa* chronic infection.

Animal models of CF are of critical importance as they contribute to our understanding of the mechanisms associated with CF disease progression and disease pathology [13]. Several CF mouse models have been characterized and reported [14]. In these mice, the *Cftr* gene is either completely knocked out or harbors mutations observed in the human population. Although human and murine CFTR have many properties in common and share 78% overall sequence identity at an amino acid level [15], some important differences in function, including channel-gating behavior, have been described [16]. CF mouse models demonstrate broad evidence of intestinal disease, but exhibit large variations in survival, anatomically confined CF ion transport defect, and general absence of CF-like lung disease. These features make them difficult to work with, especially if large numbers are required. Working with CF mice has become substantially easier after the generation of the gut-corrected CF-knockout mouse. A double-mutant CF mouse has been generated, based on the UNC-null mouse (*Cftr*^tm1Unc^) that also carries a human *CFTR* transgene controlled by rat intestinal fatty acid-binding protein gene promoter (FABP) [*Cftr*^tm1^UNCTgN(FABP*CFTR*) [17]]. FABP-mediated expression of h*CFTR* in the gut, prevents the intestinal disease observed in CF null mice. Different CF mouse models (BALB/c *Cftr* ^tm1Unc^, C57BL/6 *Cftr* ^tm1Unc^ and FVB/N *Cftr*^tm1Eur^ mice) revealed that intestinal disease phenotypes were associated with specific constituents of the intestinal microbiome. Analysis of CF mouse intestinal microbiota highlighted bacterial overgrowth and a decrease in microbiome richness and diversity [18,19,20]. Although the analysis of microbiome composition in the murine lung and gut and its dependence on CFTR mutation have been explored, the influence of *P. aeruginosa* infection has still not been investigated.

Animal models of chronic *P. aeruginosa* pneumonia have been generated to mimic the advanced stage of lung pathology in humans [14,21,22]. Chronic infection in these models is usually established by including bacterial cells in immobilizing agents where they appear to grow in microcolonies under micro-aerobic/anaerobic conditions, as they do in the mucus of CF patients. Although mild inflammatory reactions were observed with sterile agar beads, pronounced changes within the bronchi and lung parenchyma were associated with infection. Lung pathology associated with chronic experimental infection is characterized by bronchopneumonia, bronchiectasis, epithelial metaplasia, fibrosis, and alveolar exudates with inflammatory cells.

In this study, *Cftr ^tm1UNC^*TgN(FABP*CFTR*) (named KO mice) and their wild-type (WT) congenic mice were used to investigate the influence of chronic *P. aeruginosa* infection on gut and lung microbiota. Given the potential cross-talk between *P. aeruginosa* and commensal bacteria, we addressed whether changes in respective lung and gut microbiota compositions might occur during *P. aeruginosa* colonization and whether these changes are mouse model-dependent.

## 2. Results

### 2.1. P. aeruginosa Infection Influences Murine Health Both in KO and WT Mice

KO and WT congenic mice were infected with *P. aeruginosa* RP73 strain embedded in agar beads to induce chronic infection. A group of control mice were challenged with empty beads reproducing the agar-induced inflammation and stress. Both KO and WT mice infected with *P. aeruginosa* lost significantly more body weight after 2–3 days from challenge compared to empty beads, followed by a recovery at day 7 post-infection (Figure 1A). Groups of mice challenged with *P. aeruginosa* and empty beads were significantly different in body weight changes over 6 days observation, while no significance was detected comparing KO and WT mice within the group. At day 7 post-infection, no significant differences were observed between the groups—*P. aeruginosa* infected and empty beads or KO and WT mice. At day 7 post-infection, mice were sacrificed; lung, stool, and the final straight portion of the rectum, including mucosae and feces, were aseptically excised. When *P. aeruginosa* load was evaluated in the lung, the trend of CFUs increased (not significantly, *p* > 0.05) in KO mice compared to WT mice (Figure 1B).

### 2.2. Microbiota Diversity Is Influenced by CFTR Genotype in the Gut and P. aeruginosa Infection in the Lung

The 16S rRNA gene amplicon sequencing reads (Appendix A) were clustered into ASVs, and a total number of 10562 ASVs was retrieved (Appendix A). Rarefaction curves (Appendix A) indicate a satisfactory level of sample diversity (Good’s coverage, ranging from 0.99 to 1 Appendix A). Alpha diversity indices in stool and gut (Table 1) showed higher values in WT mice compared to KO mice. Of note, CFTR correction of KO mice does not rescue the gut phenotype in relation to the microbiota diversity. Lung microbiota diversity was similar in KO and WT mice, mainly due to the very small number of ASVs. In KO mice, *P. aeruginosa* infection does not influence the microbiota diversity in either stool or gut compared to the mice treated with empty beads. However, lower values of diversity in the lung microbiota occurred in KO and WT mice infected with *P. aeruginosa* compared to non-infected mice (empty beads) (*p* < 0.3). Thus, the massive colonization by the *P. aeruginosa* strain has a significant influence on lung microbiota.

Clustering of samples resulted in a strong separation of lung microbiota from the intestinal (gut and stool) microbiota, while KO and CF mice were not strongly separated (Figure 2).

The taxonomic composition of the microbiota highlighted the division between gut district (gut and stool samples) and lung, with a peculiar grouping of taxa prevalence (Figure 3). Members of the genera *Lactobacillus*, *Akkermansia*, and the family *Lachnospiraceae* were among the most abundant taxa found. The group of greatest prevalence includes many ASVs affiliated to *Lactobacillus* and *Bacteroidales.*

### 2.3. Lung Microbiota Composition Is Strongly Influenced by P. aeruginosa Infection

When gut, stool, and lung microbiota composition were analysed separately, our data showed that *P. aeruginosa* infection had a more profound impact on lung microbiota than on gut and stool microbiota (Figure 4; Table 2). In fact, most of *P. aeruginosa* infected and control mice clustered separately for lung, while less sharp clustering was detected in gut and stool (Figure 2). This evidence was further confirmed by PERMANOVA (Table 2), that indicated on the overall dataset (including both KO and WT mice, Table 2a) highly significant R^2^ values for *P. aeruginosa* infected compared to control mice in lung (*p* < 0.001, R^2^ = 0.099) and more mild differences for stool (*p* < 0.05, R^2^ = 0.03) and gut (*p* < 0.05, R^2^ = 0.04). Interestingly, on the overall dataset (irrespective of infection) WT and KO differed on gut (*p* < 0.01, R^2^ = 0.13) and stool (*p* < 0.01, R^2^ = 0.17) but not on the lung microbiota (*p* > 0.8, R^2^ = 0.01).

### 2.4. Gut Microbiota Composition of KO Mice, but Not WT Mice, Is Influenced by P. aeruginosa Infection

Further analysis on KO and WT mice separately (Table 2b) confirmed that infection by *P. aeruginosa* impacted the gut (*p* < 0.01, R^2^ = 0.17) and lung (*p* < 0.01, R^2^ = 0.16) microbiota of KO mice, but only mildly the lung microbiota of WT mice (*p* < 0.05, R^2^ = 0.08). A differential abundance analysis carried out using DeSeq2 v1.30.0 identified the ASVs/taxa differentiating the microbiota in the gut of infected mice from the CF gut control mice (adjusted *p* value < 0.05). We found 55 differentially represented ASVs (Appendix A). Most of them belonged to the *Firmicutes* phylum.

## 3. Discussion

Among the many facets of CF, the microbiological status of patients is of great interest due to the recurrent, chronic microbial infection of the airways. In addition to airway microbiome alteration, CF patients show altered fecal microbiomes, correlated with gastrointestinal inflammation and nutrient malabsorption [4]. The association between lung and gut microbiome appears to be important in the pathogenesis of lung conditions as gut microbiota has been shown to influence lung inflammatory responses in CF [3,5,23]. Until now, it has been unclear whether alterations in the gut microbial species is the cause or effect of lung diseases. Investigating how gut microbiota impact other distant organs opens up new hopes for the development of therapeutic approaches to CF respiratory infections [24]. As a matter of fact, due to the complex cross-talk, the causality between lung diseases and gut microbiota is still under explored.

In this study, we investigated the lung and gut microbiome in KO and WT mice following *P. aeruginosa* infection to examine how the murine microbiome influences *P. aeruginosa* infection and its dependence on the CFTR genotype.

Several CF mouse models are available, but all the animals have severe obstructive gut disease. Differently to the human disease, the murine gut phenotype leads to high pre- and post-weaning mortality. Thus, to exclude gut pathology as a possible confounding factor on lung phenotype, we selected the gut-corrected CF mice *Cftr ^tm1UNC^*TgN(FABP*CFTR*). We observed that, following *P. aeruginosa* infection, the bacterial loads recovered from the murine airways were higher for CF mice compared to WT mice, even if the differences were not statistically significant. The weight loss in *P. aeruginosa*-infected mice was associated with a decrease in Firmicutes phyla abundance, suggesting that *P. aeruginosa* can play a possible role in driving the biodiversity of the gut flora. These results were in agreement with other works which found that significantly reduced levels of Firmicutes correlated to weight loss in mice and humans [25,26].

Using murine models of CF infection, we compared gut and stool dysbiosis in CF mice microbiota to that of healthy WT mice with a higher microbial diversity in both gut and stool, with similar data observed for both murine models [20] and patients with CF [27]. As stated by Meeker et al. [28], CFTR dysregulation drives active selection of the gut microbiome suggesting that restoring CFTR function could favor the restoration of a healthy gut microbiota to that found in healthy, non-CF people. No difference in alpha diversity indices of the lung microbial communities was found between WT and KO mice, mainly due to the very small number of ASVs retrieved. CF mice infected with *P. aeruginosa* showed a lower diversity in lung microbiota with respect to the uninfected CF mice, most probably due to the massive colonization by *P. aeruginosa*, while no altered microbial diversity was found in the gut district. Lung microbiota composition is strongly impacted by *P. aeruginosa* infection as revealed by PCA and PERMANOVA analyses. *P. aeruginosa* infection had an effect on gut microbiota of CF mice, but not on WT mice, suggesting that the pulmonary chronic infection in CF mice may lead to altered intestinal mucosa in CF mice, irrespective of CFTR lack of expression [28]. Finding that in CF mice, gut microbiota alone (and not stool) was affected by *P. aeruginosa* raised the question of whether the mucosa-adhering microbial biofilms in the gut could be more susceptible to alteration of mucosal trophism/inflammation. We cannot exclude that, in the longer term, stool microbiota may also display the same changes. In mice, the loss of *Cftr* gene function was found to be responsible for the intestinal dysbiosis [3] and a close relationship between *Cftr* genotype and microbiome constitution was demonstrated [28]. In this study, the pulmonary chronic infection caused the alteration of intestinal mucosa in CF mice, not related to a lack of CFTR expression in the gut. We can hypothesize the existence of an “axis” from lung to gut, that is not directly related to CF defect directly, but to the inflammation status. It has been well reported that the respiratory tract has its own microbiota and lung inflammation can lead to intestinal dysbiosis [29]. Following *P. aeruginosa* infection, the differential represented by ASVs in gut microbiota, mainly belonging to Firmicutes phylum, following *P. aeruginosa* infection, underline the role of this group of bacteria with known roles in immune modulation of the gut microbiome. A recent opinion paper [30] highlighted the importance of the mouse model for the study of gut to lung axis in CF including the study of chronic infection with agar beads. However, the exact mechanisms of microorganism selection by mice genotype are unknown. Future studies should investigate the effect of *P. aeruginosa* infection on the interaction between pathogenic bacteria and intestinal microbiota. Since the host–microbe interaction includes core microbial metabolisms and virulence factors, as well as the host’s defensive countermeasures, meta’omics studies can be envisaged to fully interpret results from the *P. aeruginosa* chronic infection mouse model in a systems-biology framework [31]. Moreover, since *P. aeruginosa* strains exhibit extensive genome variation and evolution in patients’ airways [32,33], experiments with different *P. aeruginosa* strains or mutant collections would be helpful to elucidate possible bacteria genotype × host genotype interactions.

## 4. Materials and Methods

### 4.1. Ethic Statement

Animal studies adhered to the Italian Ministry of Health guidelines for the use and care of experimental animals (IACUC 908).

### 4.2. Mouse Model and P. aeruginosa Chronic Infection

Male *Cftr ^tm1UNC^*TgN(FABP*CFTR*) are bitransgenic mice harboring the FABP-h*CFTR* transgene [rat fatty acid binding protein 2, intestinal promoter directing expression of a human *CFTR* (ATP-binding cassette sub-family C, member 7) gene] and a targeted KO mutation of the *Cftr* gene.

(*Cftr ^tm1UNC^*TgN(FABPCFTR) (named KO mice) and their wild-type congenic mice (named WT mice) (12–19 weeks of age) were maintained individually in single sterile cages in specific pathogen-free conditions in a ventilated isolator for two weeks before infection to consent acclimatization and to avoid microbiome perturbations. Mice were anesthetized by an i.p injection of 2.5% Avertin (2,2,2-Tribromoethanol, Sigma-Aldrich, Merck KGaA, Germany) and intratracheally infected with 3–4 × 10^5^ of clinical strain RP73 *P. aeruginosa* [21], embedded in agar beads. A group of mice were challenged with sterile empty beads as a control. Mice were monitored daily for the healthy status and for the changes in body weight; the animals were sacrificed at seven days post-infection to track changes during long-term chronic infection as described in the experimental schedule (Figure 5). After the sacrifice, lungs, stool and the final straight portion of the rectum (mucosae + feces) were aseptically excised. Lungs were homogenized and split into aliquots for bacterial load count and microbiota analysis. An aliquot of 200 μL was immediately plated for bacterial evaluation, 500 μL was centrifuged and the supernatant was collected and frozen at −80 °C for additional analysis. In the remaining aliquot ATL buffer (15% of total volume, Qiagen, Germany) was added and stored at −80 °C for DNA extraction and metagenomic investigations. Stool and rectum were frozen at −80 °C to evaluate the composition of gut microbiome. A total of four experiments were performed. Statistical analyses were performed with GraphPad Prism (GraphPad Software, Inc., San Diego, CA, USA) using a two-way ANOVA with Bonferroni’s multiple comparison test for body weight changes and Mann Whitney *t*-test for the other readouts. Outlier data, identified by Grubbs’ test, were excluded from the analysis.

### 4.3. DNA Extraction and Sequencing

To set-up the best methodology of DNA extraction from mice tissues (lung homogenates, gut, and stool) stored at −80 °C, different methods were considered. After some preliminary tests, DNeasy PowerSoil Pro Kit (Qiagen, Germany) was selected for the stool and gut samples. Frozen stool and gut samples stored at −80 °C were added to PowerBead Pro tubes and treated as described in the manufacturer’s instructions. DNA was recovered in 10mM Tris pH 7.4 and quantified using Qubit 2.0 (Thermo Fisher Scientific, MA, USA) fluorometer as per manufacturer’s instructions. For lung homogenates, samples were subjected to DNA extraction by using a Qiagen DNeasy Blood and Tissue Kit, following the procedure reported by [34] with minor modification. Samples were subjected to bead beating for 1 min in DNA isolation bead tubes (Qiagen, Germantown, MD, USA) prior to kit use. The DNeasy Blood and Tissue protocol was modified to use 40 μL proteinase K instead of the recommended 20 μL, and samples were eluted with 50 μL of buffer AE instead of the suggested 200 μL. A total of 150 samples were collected (Appendix A). The V3–V4 hypervariable region (approximately 469 bp) of the 16S rRNA gene, which has been used for taxonomic classification of the microbial community in human microbiome studies, was amplified. DNA was then sequenced according with the MiSeq platform (Illumina). Sequences are deposited under SRA PRJNA700486.

### 4.4. Bioinformatic and Statistical Analyses

The DADA2 pipeline (version 1.10.0) was used to reconstruct amplicon sequence variants (ASVs) from Illumina reads. Both ASV reconstruction and statistical analyses were performed in the R environment version 3.4.3 (http://www.R-project.org accessed on 1 September 2021). For a complete description of all the steps performed see the Appendix A section. Briefly, primers used for V3-V4 amplification were detected and removed using cutadapt version 1.15. Low quality reads were discarded using the filterAndTrim function with an expected error threshold of two for both forward and reverse read pairs. Denoising was performed using the dada function after error rate modelling (learnErrors function). Denoised reads were then merged discarding those with any mismatches and/or an overlap length shorter than 20bp (‘mergePairs’ function). Chimeric sequences were removed using the removeBimeraDenovo function. Taxonomical classification was performed using DECIPHER package version 2.10.2 against the latest version of the pre-formatted Silva small-subunit reference database (SSU version 132. Available online: http://www2.decipher.codes/Downloads.html (accessed on 23 November 2020)). All sequences classified as chloroplasts, mitochondria, Archaea and Eukarya were removed. Bacterial diversity in each sample was computed using the inverse Simpson index as implemented in the diversity function of the vegan package. Distances across samples were reported using non-metric multidimensional scaling (nMDS) as implemented in the metaMDS function of the vegan package, with 300 random starts and monotone regression. Differential abundance analysis was performed using the R package DESeq2 v1.30.0 [35].

## 5. Conclusions

Overall, our finding indicates that the *P. aeruginosa* infection influences murine health both in KO and WT mice significantly changing body weight both in CF and WT mice. The pulmonary chronic infection in CF mice may lead to altered intestinal mucosa not directly related to the CFTR genotype. Our results reinforce the hypothesis of a cross-talk between lung and gut microbiota in relation to *P. aeruginosa* chronic infection and CFTR mutation and emphasize the need of animal models to mechanistically decipher the influence of infection on whole host physiology. Results from such models, though oversimplified with respect to human patients, can give us a set of tools to unlock the potential of microbiome-based personalized medicine in major disease areas including CF. Moreover, such animal models will also assist in the development of therapeutic treatments which while targeting a specific microbiome (e.g., the gut microbiome) produce pleiotropic effect on other relevant organs (e.g., lung) as we may envisage from the presented results on a gut-to-lung axis in CF mice.

## Figures and Tables

**Figure 1 ijms-22-12169-f001:**
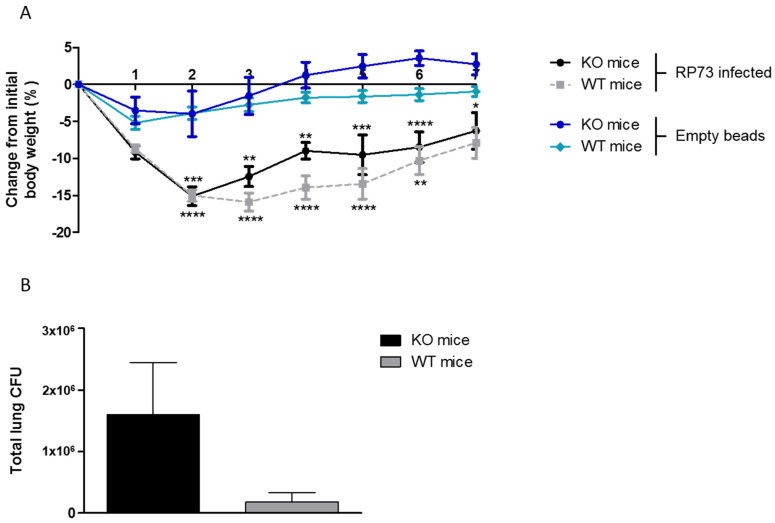
Body weight changes and *P. aeruginosa* load in KO and WT congenic mice. KO male and WT male mice were infected with 3–4 × 10^5^ CFU of *P. aeruginosa* RP73 embedded in agar beads by intratracheal inoculation. A control group was challenged with empty beads. Mice were weighed daily, and the percentage change from the initial body weight was averaged for each group of mice (**A**). After day 7 post-inoculation, the mice were sacrificed, lungs were excised, homogenized and plated onto tryptic soy agar to determine the bacterial load (**B**). Data are presented as mean ± SEM. The data were pooled from four independent experiments (RP73 infected: KO mice *n* = 13 and WT mice *n* = 17; empty beads: KO mice *n* = 7 and WT mice *n* = 8). * *p* < 0.05, ** *p* < 0.01, *** *p* < 0.001, **** *p* < 0.0001. Comparisons between KO *P. aeruginosa* infected vs. KO mice challenged with empty beads and WT *P. aeruginosa* infected vs. WT mice challenged with empty beads were performed by two-way ANOVA with Bonferroni’s post-test.

**Figure 2 ijms-22-12169-f002:**
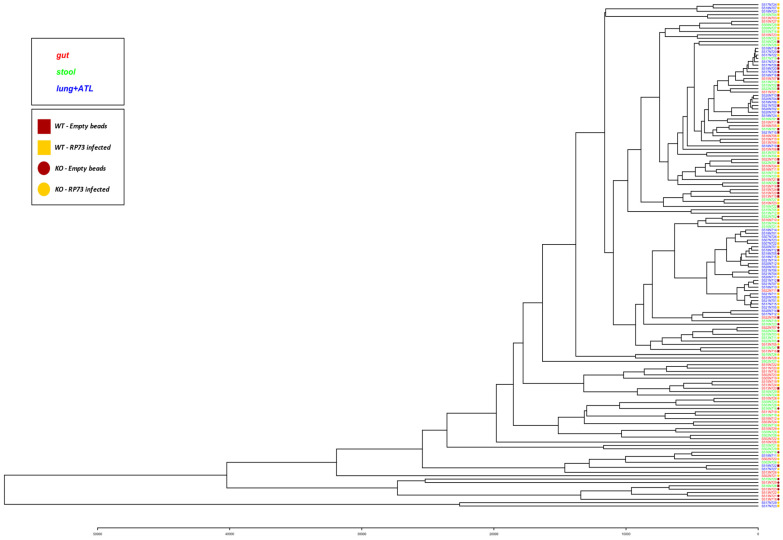
Clustering of mice microbiota. Codes of samples are as those reported in Appendix A.

**Figure 3 ijms-22-12169-f003:**
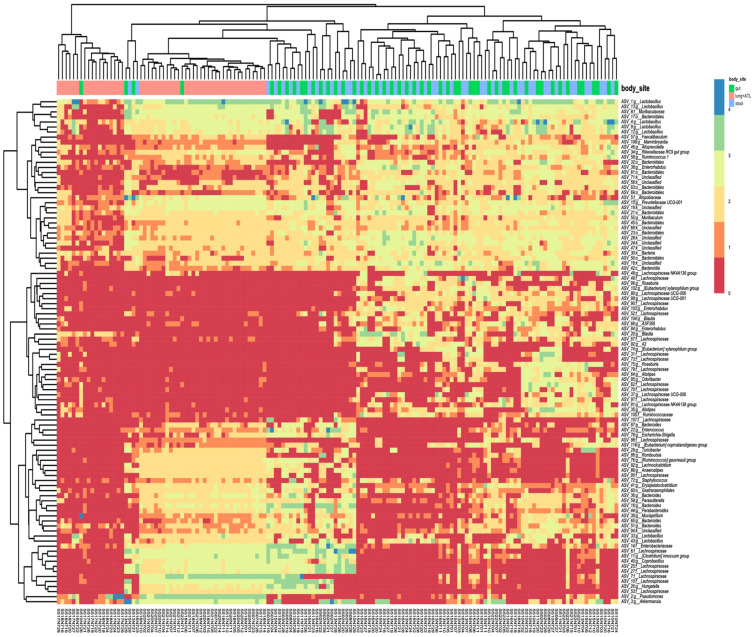
Taxonomic composition and clustering of samples based on the most abundant ASV (top 100). The heat-map showing the abundance of the top 100 ASVs in samples is reported.

**Figure 4 ijms-22-12169-f004:**
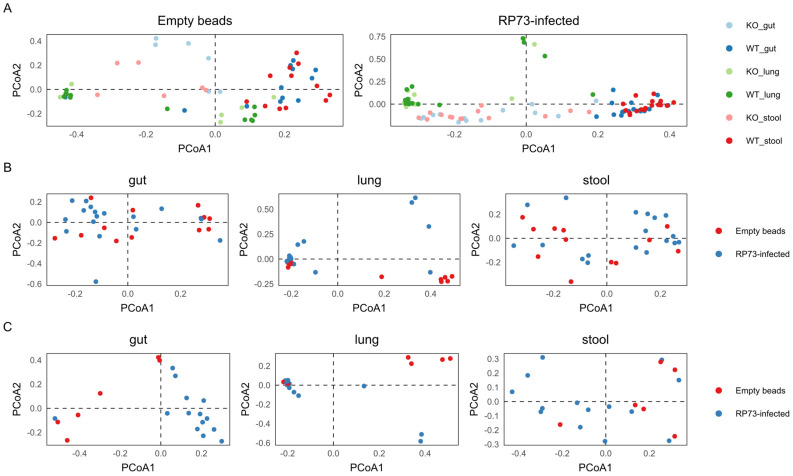
Principal component analysis (PCA) of mice microbiota. Pales shows the distribution of samples with respect to (**A**) infections; (**B**) WT mice and (**C**) KO mice in relation to districts. WT mice: empty beads *n* = 8 and RP73 infected *n* = 17; KO mice: empty beads *n* = 7 and RP73 infected *n* = 13.

**Figure 5 ijms-22-12169-f005:**
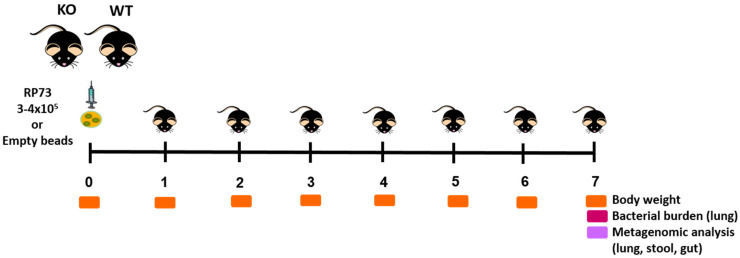
Experimental design. Mice were intratracheally infected with 3–4 × 10^5^ of clinical strain *P. aeruginosa* RP73 embedded in agar beads or empty beads and were sacrificed after seven days. Lungs, stool, and the final straight portion of the rectum (mucosae + feces) were aseptically excised and processed for metagenomic analysis. Read-outs of the disease progression were body weight changes, health status and CFUs in the lung.

**Table 1 ijms-22-12169-t001:** Alpha diversity of mice microbiota. WT and KO mice not infected (empty beads) and infected with *P. aeruginosa* (RP73) are shown (WT mice: empty beads *n* = 8 and RP73 infected *n* = 17; KO mice: empty beads *n* = 7 and RP73 infected *n* = 13). Values (means ± standard deviation) of the Inverse Simpson index are reported.

	Gut	Stool	Lung
WT (empty beads)	37.26 ± 12.78	50.43 ± 24.48	33.45 ± 31.63
WT (RP73 infected)	41.52 ± 20.21	30.88 ± 21.48	15.02 ± 7.81
KO (empty beads)	9.81 ± 10.93	16.73 ± 10.69	40.69 ± 28.24
KO (RP73 infected)	19.12 ± 9.93	18.44 ± 5.79	15.21 ± 7.90

**Table 2 ijms-22-12169-t002:** Infection significantly impacts on both lung and gut microbiota. Impact of body site, genotype and infection on the overall dataset (a); or restricted to RP73 infected compared to controls (b). Results of Permanova on the microbiota composition evaluating the differences among conditions and genotypes. The probability (Pr > F after F statistic calculation) and R^2^ values are reported for each body site (gut, stool, lung). WT mice: empty beads *n* = 8 and RP73 infected *n* = 17; KO mice: empty beads *n* = 7 and RP73 infected *n* = 13. Asterisks indicate significant values at *p*-values <0.05 *, 0.01 **, 0.001 ***.

		Gut		Stool		Lung	
		Pr(>F)	R2	Pr(>F)	R2	Pr(>F)	R2
(a)	Condition (RP73 infected—control)	0.006 **	0.04305	0.026 *	0.03572	0.001 ***	0.09996
	Genotype (KO—WT)	0.001 ***	0.13225	0.001 ***	0.17754	0.894	0.01035
(b)	WT (RP73 infected—control)	0.098	0.04679	0.097	0.0461	0.023 *	0.08876
	KO (RP73 infected—control)	0.002 **	0.16172	0.100	0.0819	0.007 **	0.17221

## Data Availability

Sequences reads are deposited at the National Center for Biotechnology Information (NCBI) database under the Bioproject PRJNA700486.

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
