# Peer review of "Lung and Gut Microbiota Changes Associated with Pseudomonas aeruginosa Infection in Mouse Models of Cystic Fibrosis"

_ijms, 2021, doi:10.3390/ijms222212169_

Round 1

Reviewer 1 Report

1) Some sentences need to be rephrased and thus, be more clearly stated. For example, in the Introduction, line 57 should be rephrased as follows: "Although evidence that CFTR mutations have an effect on gut microbiota both in humans and CF mice [13-14], it is not clear how this gut-lung dysbiosis is influenced by Pseudomonas aeruginosa chronic infection.

2) Please mention the number of mice used per category per experiment.

Author Response

We thank the reviewer for the constructive comments and suggestions. We have carefully considered them and changes have been done accordingly. In reply to your major and minor comments, specific Answers are reported below.

Q1. Some sentences need to be rephrased and thus, be more clearly stated. For example, in the Introduction, line 57 should be rephrased as follows: "Although evidence that CFTR mutations have an effect on gut microbiota both in humans and CF mice [13-14], it is not clear how this gut- lung dysbiosis is influenced by Pseudomonas aeruginosa chronic infection.

R1. We modified the manuscript according to reviewer’s suggestion. The manuscript has been extensively revised by a native English speaker following the Editor’s suggestion.

Q2. Please mention the number of mice used per category per experiment.

R2. We have included the number of mice used for each experiment.

Reviewer 2 Report

I review this article with great interest. below are my comments:

1.- I think more details about the model are needed. particular in terms of growth time (age of the mice when experiments were perform) and more important evidence or not of lung disease at the time of the pseudomonas infection.

2.- the discussion and conclusion can be edited to be more clear and deliver their point and finding. it can also discuss the potential effect of CFTR modifiers on their findings.

Author Response

Thank you very much for your careful reading and useful comments. Based on your comments, we have carefully revised our manuscript, re-writing some parts and expanding some explanations. We hope we have addressed your concerns. Detailed reply to your comments:

Q1. 1.- I think more details about the model are needed. In particular in terms of growth time (age of the mice when experiments were performed) and more important evidence or not of lung disease at the time of the pseudomonas infection.

R1. We have included more details on the animal model of infection in the Introduction lines 85-93. The ages of the mice included in the experiment are reported in the Materials and Methods line 258 page 8.

Q2. 2.- the discussion and conclusion can be edited to be more clear and deliver their point and finding. It can also discuss the potential effect of CFTR modifiers on their findings.

R2. We thank the Reviewer for this valuable suggestion. Accordingly, we modified the Discussion (lines 206-210 page 7, and 244-250 page 8) and improved the conclusion (lines 326-338). We cannot discuss the potential effect of CFTR modifiers since we are using only one murine background strain and not comparing mice from different backgrounds.

Round 2

Reviewer 2 Report

Appreciate review and responses.